# Genetic variants in the leptin-melanocortin pathway and their joint effects with physical activity and sleep duration on risk of childhood obesity

Xiaotong Liang[1], Shun Pan[1], Jiting Ji[1], Yuying Deng[1], Miao Chen[1], Ziwei Huang[1], Jiaru Deng[1], Min Chen[1], Tiantian Xiong[2], Jianping Liang[3]*, Li Liu [iD][1]*

1 Department of Epidemiology and Biostatistics, School of Public Health, Guangdong Pharmaceutical University, Guangzhou, China, 2 Department of Occupational Health, Shenzhen Bao'an District Public Health Center, Shenzhen, China, 3 Department of School Health, Guangzhou Health Care Promotion Center for Primary and Middle Schools, Guangzhou, China

* 413590176@qq.com (JL); pupuliu919@163.com (LL)

## Abstract

### Objective

This study aimed to investigate the associations between functional genetic variants in leptin-melanocortin pathway and risk of childhood obesity, as well as exploring gene-lifestyle interactions on obesity risk.

### Methods

A frequency-matched case-control study included 1123 children and adolescents with obesity and 1231 controls from Guangzhou, China. Twelve potentially functional genetic variants were genotyped. Multivariate logistic regression and classification and regression tree were applied to identify obesity-associated genetic variants. Unweighted and weighted genetic risk scores (GRSs) were constructed, and their interactions with physical activity and sleep were assessed using relative excess risk of interaction (*RERI*), attributable proportion of interaction (*AP*) and the interaction of odds ratio (*IOR*).

### Results

We identified important genetic variants associated with obesity: *MC4R* rs17782313, rs12970134, *LEPR* rs1137101, and *POMC* rs6713532. High unweighted GRS was associated with a higher obesity risk than low GRS (*OR* = 1.40, 95% *CI* = 1.09–1.79), with similar results for the weighted GRS. High unweighted GRS combined with inadequate sleep or physical activity conferred increased risk of obesity compared to low/medium GRS with healthy lifestyle (*OR* = 1.70, 95% *CI* = 1.28–2.26; *OR* = 1.59, 95% *CI* = 1.17–2.17). However, the 95% *CI*s for all *RERI*s and *AP*s contained 0, and for *IOR*s contained 1, suggesting no significant interaction.

**Data availability statement:** The participants' demographic and lifestyle dataset is publicly available in Figshare (DOI: https://doi.org/10.6084/m9.figshare.32085195). Individual genetic data are not publicly available due to participant privacy protection, ethical restrictions, and the Regulation on the Administration of Human Genetic Resources of China. Access to individual genetic data may be granted to other researchers only after approval from the Ethics Committee of Guangdong Pharmaceutical University (contact: gylun-li@163.net). The original statistical analysis code and programming log are provided as supplementary materials (S1 Text).

**Funding:** This work was supported by the Natural Science Foundation of Guangdong (2023A1515010105).The funders had no role in study design, data collection and analysis, decision to publish, or preparation of the manuscript.

**Competing interests:** The authors have declared that no competing interests exist.

## Conclusions

Four genetic variants in the leptin-melanocortin pathway are associated with the risk of obesity, both individually and jointly. Furthermore, the joint effect of these genetic variants with inadequate sleep and physical activity may contribute to obesity development in children and adolescents.

## Introduction

Obesity is a global epidemic that is continually advancing [1]. In China, there is also a rapidly increasing trend of obesity among children and adolescents, with the prevalence projected to reach 15.1% in 2030 [2]. Childhood obesity is associated with adverse social and health outcomes for children and adolescents, as well as an increased likelihood of adult obesity and serious related conditions, such as cardiometabolic disease and certain cancers [3]. Obesity arises from an intricate interplay between genetic and environmental risk profiles [4], with genetic component accounting for 40–70% of inter-individual variation [5]. Consequently, uncovering the genetic mechanisms of obesity is imperative for the development of targeted interventions to combat obesity.

Early research on monogenetic obesity identified the leptin-melanocortin pathway as a key regulator of energy homeostasis, with disruptions leading to severe obesity [6]. The involvement of multiple genes within this pathway in polygenic obesity were subsequently validated by candidate gene studies and genome-wide association studies (GWASs), including leptin (*LEP*), leptin receptor (*LEPR*), pro-opiomelanocortin (*POMC*), melanocortin-4 receptor (*MC4R*), and neuropeptide Y (*NPY*) [7,8]. The leptin-melanocortin pathway begins with LEP binding to its receptors, activating POMC neurons while inhibiting NPY neurons (Fig 1) [9]. Activated POMC neurons release α-melanocyte-stimulating hormone (αMSH) which stimulates melanocortin 4 receptor (MC4R) and MC3R, suppressing food intake and promoting energy expenditure [10,11]. Although NPY and POMC neurons interact with the same populations of MC3R and MC4R, they exert opposing effects, with NPY neurons acting as the primary drivers of feeding behavior [12].

Genetic variants within the leptin-melanocortin pathway have been associated with a predisposition to obesity [13]. Pathogenic mutations in *MC4R* have been identified in up to 5% of cases of severe childhood obesity, while additional genetic variants, including *LEP* rs10487505, *LEPR* rs11208659, *MC4R* rs17782313, and *POMC* rs6545975, have also been implicated in severe and early-onset obesity [7]. Both monogenic and polygenic forms of obesity share a common underlying biological basis. Recent GWASs have continued to link genetic variants such as rs17782313, rs11208659 and rs6545975 to both common and severe obesity [7]. To date, however, there is a paucity of comprehensive genetic association investigations focusing on the core leptin-melanocortin pathway in relation to childhood obesity.

Given the multifaceted nature of the etiology of obesity, exploration of the cumulative and interactive effects of genetics along with environmental exposure could

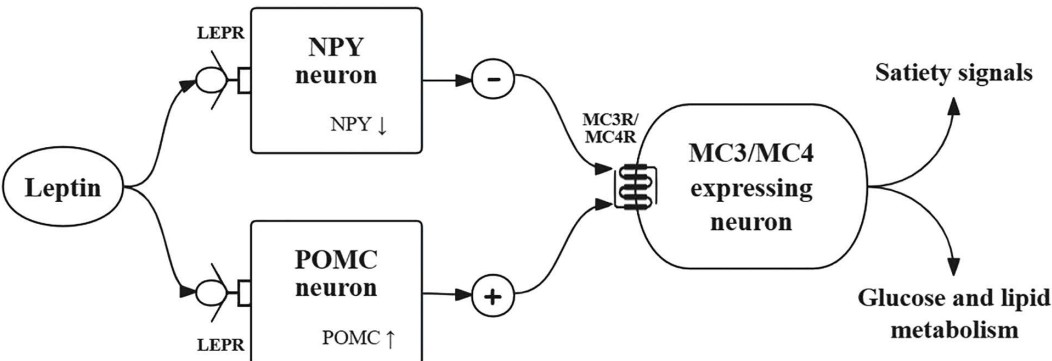

**Fig 1. The leptin–melanocortin pathway.** Abbreviations: LEPR, leptin receptor; NPY, neuropeptide Y; POMC, pro-opiomelanocortin; MC3R, melanocortin-3 receptor; MC4R, melanocortin-4 receptor.

enhance our understanding of the complexity of the obesity epidemic. Exercise and sufficient sleep could improve leptin resistance and promote leptin signaling [14]. Current studies have mainly concentrated on how single genetic variants in the leptin pathway interact with some behavioral factors, such as *MC4R* rs12970134 in interaction with physical activity [15], or the interaction between leptin-related polygenic risk and short sleep duration in modifying childhood obesity susceptibility [16]. Despite these existing findings, the joint effect of multiple functional variants across this pathway and their interaction with physical activity and sleep remains largely uncharted, representing a critical knowledge gap for unraveling the full complexity of obesity etiology.

To fill this gap in knowledge, we adopted a pathway-driven approach that aggregated information from functional genetic variants, and examined how genetic risk interacted with major modifiable lifestyle factors. This study enrolled 1123 children and adolescents with obesity and 1231 controls from Guangzhou, China, using a frequency-matched case-control study design. In this study, we investigated the genetic associations between childhood obesity and 12 potentially functional genetic variants within the core genes of the leptin-melanocortin pathway, including *LEP*, *LEPR*, *POMC*, *NPY*, *MC3R* and *MC4R*. We further calculated the genetic risk scores (GRSs) for childhood obesity, and particularly focused on the potential interaction between the GRSs with physical activity and sleep duration. This work sheds new light on both the etiology of obesity and potential prevention strategies to address childhood obesity in a more precise manner.

## Materials and methods

### Study populations

This frequency-matched case-control study was conducted in Guangzhou, the capital city with the largest population in the south of China. Between June 2016 and July 2017, participants were drawn from the physical monitoring program at Guangzhou Health Care Promotion Center, including students aged 7−8 from 11 primary schools, 11−14 from 26 junior schools, and 14−18 from 13 senior high schools. Obesity was defined as a body mass index (BMI) greater than the 95th percentile for sex and age, and normal weight as BMI below the 85th percentile, according to the BMI reference norm for screening overweight and obesity in Chinese children and adolescents issued by the Working Group on Obesity in China [17]. A total of 1123 children and adolescents with obesity (213 children aged 7−8 years and 910 aged 11−18 years) were recruited as cases, along with 1231 children and adolescents with normal weight as controls. Controls were frequency-matched to cases by sex, age, and school. All participants were unrelated Han Chinese children and adolescents. The inclusion criteria were children and adolescents in good psychophysical condition, without chronic or acute pathological condition, who consented to participate in this study. This study was approved by the Ethics Committee of Guangdong

Pharmaceutical University (Med Ref No. 2016–17, Med Ref No. 2018−27). Written informed consent was obtained from the parents or legal guardians of each participant prior to their inclusion in the study.

## Anthropometric measurements

Height and weight were measured by trained technicians using standardized protocols and calibrated equipment. Height was measured to the nearest 0.1 cm using a mechanical height gauge, and weight was measured to the nearest 0.1 kg using a lever scale. For each participant, both measurements were done twice in light clothing without shoes, and the average of each measurement was calculated. BMI was calculated as weight (kg) divided by height (m) squared.

## Other data collection

At study entry, epidemiological data were collected using a self-administered questionnaire named *Guangzhou Primary and Secondary School Students' Physical Health Status Questionnaire*. This questionnaire was meticulously designed by our project team members and subsequently refined by experts, and a pilot survey was conducted, with a Cronbach's α coefficient of 0.874. The questionnaire was structured into two sections, one to be completed by the children and the other by their parents. The children's section covered demographic characteristics and lifestyle behaviors (including physical activity and sleep duration). The parental section mainly encompassed information about family characteristics, such as the parents' education levels and household income per capita. For children in primary school grade 2, the questionnaire section was completed with the assistance of their parents, while secondary school students completed the questionnaire independently. Trained investigators administrated the questionnaire in class and provided uniform guidance on how to fill out the questionnaire to children. Children were instructed to hand the parent-completed section to their parents and return the completed questionnaire to school.

In terms of the children's lifestyle behaviors, physical activity status was classified as adequate (≥1 hour/day) or inadequate (<1 hour/day), based on the Physical Activity Guidelines for Chinese children and adolescents [18]. For sleep duration, we applied Chinese national standards which recommend: ≥10 hours daily for primary students, ≥9 hours for junior high students, and ≥8 hours for senior high students [19]. According to these cutoff points, sleep duration was defined as adequate sleep or inadequate sleep.

## Genetic variants selection and genotyping

Potentially functional genetic variants in the *LEP*, *LEPR*, *POMC*, *NPY*, *MC3R*, and *MC4R* were screened using a comprehensive strategy integrating multiple bioinformatical databases. Initially, a catalog of genetic variants spanning from 500-bp upstream to 500-bp downstream of the candidate genes in the populations of Han Chinese South (CHS) and Han Chinese in Beijing, China (CHB) was derived from the 1000 Genomes Project (https://www.internationalgenome.org) and the Chinese Millionome Database (http://cmdb.bgi.com). Subsequently, the genetic variants with a minor allele frequency (MAF) less than 0.05 were filtered out. Next, genetic variants located in the 5'-flanking regions, 5'-untranslated regions, 3'-untranslated regions, and exons were identified using the Ensembl database (https://grch37.ensembl.org/index.html). Functional annotations of these genetic variants were then performed based on biological knowledge from published research or through online bioinformatical tools including SNPinfo, HaploReg, and RegulomeDB for non-coding regions, and SIFT, Polyphen2, and CADD for coding regions. The inclusion criteria for candidate genetic variants were: 1) MAF > 0.05 in the Chinese Han population; 2) putative functional genetic variants, possibly affecting transcription activity, alternative exon splicing or encoding critical amino acid substitutions; 3) genetic variants in low linkage disequilibrium with each other ($r^2 < 0.8$). Finally, 12 genetic variants were selected for genotyping: rs1349419 and rs2167270 in *LEP*, rs11208659, rs1137100 and rs1137101 in *LEPR*, rs6713532 in *POMC*, rs16141 in *NPY*, rs6127698 and rs3746619 in *MC3R*, and rs17782313, rs12970134, and rs8087522 in *MC4R* (S1 and S2 Tables).

A whole blood sample (~2mL) was collected from all cases and controls and stored at −80°C until DNA isolation. Genomic DNA was isolated from the white blood cells using the OMEGA DNA Blood Kit (D3471-02; Omega Bio-Tek Inc.) [20]. Genotyping was performed using the TaqMan PCR assay in an ABI 7900HT Fast Real Time PCR System (Applied Biosystems). Each amplification was carried out in a 5 μL reaction system, with the universal cycling conditions. Genotyping was performed blindly, without knowledge of case or control status. We also repeatedly genotyped 2% randomly selected samples, with a concordance rate of 100%. The call rates of genotyping were all above 99% (S1 Table).

## Statistical analysis

A Chi-square test was used to compare categorized variables and a *t*-test for continuous variables between cases and controls. The Hardy-Weinberg equilibrium (HWE) of the genotypic distribution was assessed in controls using a Chi-square goodness-of-fit test with 1000 permutations for multiple comparison corrections. A multivariate logistic regression model was applied to estimate genetic associations with obesity risk, adjusting for age, sex, maternal and paternal education levels, and household incomes. A 1000 times permutation test was conducted for correction of multiple comparisons in the genetic association analyses [21]. In parallel, classification and regression tree (CART) analysis, a non-parametric analytical method, was conducted to screen out additionally important genetic variants for obesity [22]. The Gini impurity index was adopted as a splitting criterion for growing a tree, with a minimum of 100 cases for each Parent Node and 50 for each Child Node. Subsequently, 10-fold cross-validation was applied to evaluate the predictive accuracy of the tree. The risk for each subgroup was then evaluated by comparison with the subgroup with the lowest proportion of cases, using logistic regression with the same adjustment set as mentioned above. Finally, genetic variants contained in the highest risk subgroup of CART, together with those showing significant associations in traditional logistic regression analysis, were considered as important genetic variants for obesity predisposition.

The GRS was calculated as the sum of risk alleles counts (unweighted GRS) or the sum of risk alleles carried by each individual weighted by the effect size of each variant (weighted GRS) [23], incorporating the important genetic variants. Both GRSs were then categorized into low, medium, and high groups using the first and third quartiles as the cut-off points. *OR*s (odds ratio) and their 95% *CI*s (confidence interval) for obesity risk were calculated for the individuals with medium or high GRS in the multivariate logistic regression model, treating the low GRS group as a reference.

Genetic association analyses of the GRSs were further stratified separately by sex and age group, with heterogeneity between regression coefficients tested by the *Z* test. Based on the crossover analysis, the relative excess risk of interaction (*RERI*) and the attributable proportion of interaction (*AP*) were calculated to characterize the potential interaction between physical activity, sleep, and GRSs on the additive scale. The multiplicative interaction was also evaluated using the interaction of *OR* (*IOR*) [24]. Their 95% *CI*s were estimated using the bootstrap method with 1000 replications [25], with *RERI* and *AP* unequal to 0 and *IOR* unequal to 1 indicating significance. All statistical analyses were conducted in SAS 9.4 and R v4.2.3. Two-tailed test was used, and $P < 0.05$ was regarded as being statistically significant. The statistical power of this study was estimated using PASS 2021 software. When the frequency of a risk allele was 0.1, this study had a statistical power of 88.8% to detect an *OR* of 1.5.

## Results

### Characteristics of cases and controls

The characteristics of the 1123 cases and 1231 controls are summarized in Table 1. Cases and controls were comparable in terms of age, sex, grade, education level of parents, and household incomes (all $P > 0.05$). The case-control differences in lifestyle factors, including physical activity and sleep duration, were not statistically significant (all $P > 0.05$).

**Table 1. Participants' characteristics.**

| Characteristics | Cases N = 1123 | Controls N = 1231 | $\chi^2/t$ | P |
|---|---|---|---|---|
| Age groups | | | 0.01 | 0.997 |
| 7-8 years | 213 (19.0) | 232 (18.8) | | |
| 11-14 years | 548 (48.8) | 601 (48.8) | | |
| 15-18 years | 362 (32.2) | 398 (32.3) | | |
| Grades | | | 0.01 | 0.995 |
| Primary school, grade 2 | 213 (19.0) | 232 (18.8) | | |
| Junior school, grade 1 | 546 (48.6) | 598 (48.6) | | |
| Senior school, grade 1 | 364 (32.4) | 401 (32.6) | | |
| Sex | | | <0.01 | 0.952 |
| Boys | 765 (68.1) | 840 (68.2) | | |
| Girls | 358 (31.9) | 391 (31.8) | | |
| Maternal education level | | | 0.05 | 0.817 |
| High school and below | 542 (48.3) | 600 (48.7) | | |
| Junior college degree or above | 581 (51.7) | 631 (51.3) | | |
| Paternal education level | | | 1.51 | 0.219 |
| High school and below | 498 (44.3) | 577 (46.9) | | |
| Junior college degree or above | 625 (55.7) | 654 (53.1) | | |
| Household incomes per capita | | | 0.86 | 0.649 |
| <5000 Yuan/mo | 324 (28.9) | 374 (30.4) | | |
| 5000-9999 Yuan/mo | 420 (37.4) | 460 (37.4) | | |
| ≥10000 Yuan/mo | 379 (33.7) | 397 (32.3) | | |
| Height, cm | 159.9 ± 14.7 | 157.1 ± 15.9 | −4.33 | <0.001 |
| Weight, kg | 71.7 ± 19.7 | 46.1 ± 12.5 | −37.33 | <0.001 |
| BMI, kg/m$^2$ | 27.4 ± 3.8 | 18.2 ± 2.2 | −70.28 | <0.001 |
| Physical activity[a] | | | 0.56 | 0.453 |
| Adequate | 144 (13.0) | 171 (14.1) | | |
| Inadequate | 964 (87.0) | 1045 (85.9) | | |
| Sleep duration[a] | | | 2.02 | 0.156 |
| Adequate | 204 (18.4) | 252 (20.8) | | |
| Inadequate | 902 (81.6) | 960 (79.2) | | |

BMI, body mass index. Data are presented as N (%) for categorical variables, mean ± SD for continuous variables. Chi-square test was used to compare categorized variables and t-test for continuous variables.

[a]Variables with missing data.

## Associations between genetic variants in the leptin-melanocortin pathway and obesity risk

Most genetic variants in controls were confirmed to follow HWE, with the exception of rs3746619 in *MC3R*, which showed a marginally significant departure after permutation correction ($P_{permutation}$ = 0.046, S1 Table). The associations between genetic variants and obesity risk are shown in Table 2, S3 and S4 Tables. In single-locus analysis, significant differences in genotypic distribution between cases and controls were observed for rs17782313 and rs12970134 in *MC4R* (all $P < 0.05$). Carriers of the rs17782313 CT genotype had an increased risk of obesity compared with those with the TT genotype, after adjusting for age, sex, maternal and paternal education levels, and household incomes (adjusted *OR* = 1.41, 95% *CI* = 1.17–1.68, Table 2). This genetic variant was also associated with a higher risk of obesity under the dominant and additive models, with adjusted *OR*s of 1.40 (95% *CI* = 1.18–1.67) and 1.31 (95% *CI* = 1.13–1.52), respectively.

**Table 2. Association between genetic variants in *MC4R* and obesity risk in children and adolescents.**

| Genotypes | OR (95% CI) | P | OR (95% CI)[b] | P[b] | P permutation[c] |
|---|---|---|---|---|---|
| rs17782313 | | | | | |
| TT | 1.00 | | 1.00 | | |
| CT | 1.41 (1.17-1.68) | <0.001 | 1.41 (1.17-1.68) | <0.001 | <0.001 |
| CC | 1.37 (0.89-2.12) | 0.150 | 1.36 (0.89-2.10) | 0.160 | 0.152 |
| Dominant model | 1.40 (1.18-1.67) | <0.001 | 1.40 (1.18-1.67) | <0.001 | <0.001 |
| Recessive model | 1.24 (0.81-1.90) | 0.326 | 1.23 (0.80-1.89) | 0.342 | 0.331 |
| Additive model | 1.31 (1.13-1.52) | <0.001 | 1.31 (1.13-1.52) | <0.001 | 0.001 |
| rs12970134[a] | | | | | |
| GG | 1.00 | | 1.00 | | |
| AG | 1.25 (1.05-1.50) | 0.015 | 1.26 (1.05-1.51) | 0.014 | 0.021 |
| AA | 1.49 (0.97-2.31) | 0.072 | 1.49 (0.96-2.31) | 0.075 | 0.064 |
| Dominant model | 1.28 (1.08-1.52) | 0.006 | 1.28 (1.08-1.53) | 0.005 | 0.009 |
| Recessive model | 1.40 (0.91-2.16) | 0.129 | 1.39 (0.90-2.15) | 0.133 | 0.126 |
| Additive model | 1.24 (1.07-1.44) | 0.004 | 1.24 (1.07-1.44) | 0.004 | 0.008 |

*CI*, confidence interval; *OR*, odds ratio.

[a]rs12970134 data were missing for one case and three controls.

[b]Multivariate logistic regression models were adjusted for age, sex, maternal and paternal education levels, and household incomes.

[c]*P* for permutation test with 1000 times in the adjusted models.

Furthermore, the genetic association of rs17782313 remained significant even after permutation correction for multiple comparisons (both $P_{permutation} < 0.05$ for dominant and additive models).

With confounders adjusted for, similar results were observed for another genetic locus in *MC4R*, rs12970134, showing a significant association with increased obesity risk for the AG genotype compared with the GG genotype (adjusted *OR* = 1.26, 95% *CI* = 1.05–1.51, $P_{permutation} = 0.021$), as well as in the dominant model (adjusted *OR*=1.28, 95% *CI* = 1.08–1.53, $P_{permutation} = 0.009$) and additive models (adjusted *OR* = 1.24, 95% *CI* = 1.07–1.44, $P_{permutation} = 0.008$). No significant associations were found for other genetic variants (S4 Table).

## CART analysis of genetic variants in the leptin-melanocortin pathway on obesity risk

In CART, there were 8 terminal nodes constructed from interaction among 6 genetic variants, including *MC4R* rs17782313, *LEP* rs1349419, *POMC* rs6713532, *MC4R* rs8087522, *LEPR* rs1137101, and *NPY* rs16141 (Fig 2). Individuals in node 11 exhibited the highest proportion of cases (56.9%). Using individuals in terminal node 12 as the reference, individuals in terminal node 11 carrying the rs17782313 CT/CC, rs6713532 CC, and rs1137101 AG/AA genotypes showed a borderline-significant *OR* of 1.82 (95% *CI* = 0.96–3.45, *P* = 0.069, S5 Table). Additionally, the terminal node 7 associated with higher obesity risk also included the rs17782313 and rs6713532 variants. Despite not achieving statistical significance in traditional single-locus analysis, *LEPR* rs1137101 and *POMC* rs6713532 variants, were considered potentially important for obesity.

## Cumulative effect of important genetic variants on obesity

The unweighted and weighted GRSs were calculated based on four important genetic variants, *MC4R* rs17782313 (risk allele C), rs12970134 (risk allele A), *LEPR* rs1137101 (risk allele A), and *POMC* rs6713532 (risk allele T). The unweighted GRS ranged from 0 to 6, yielding a trend of increasing risk with the more risk alleles that individuals harbored

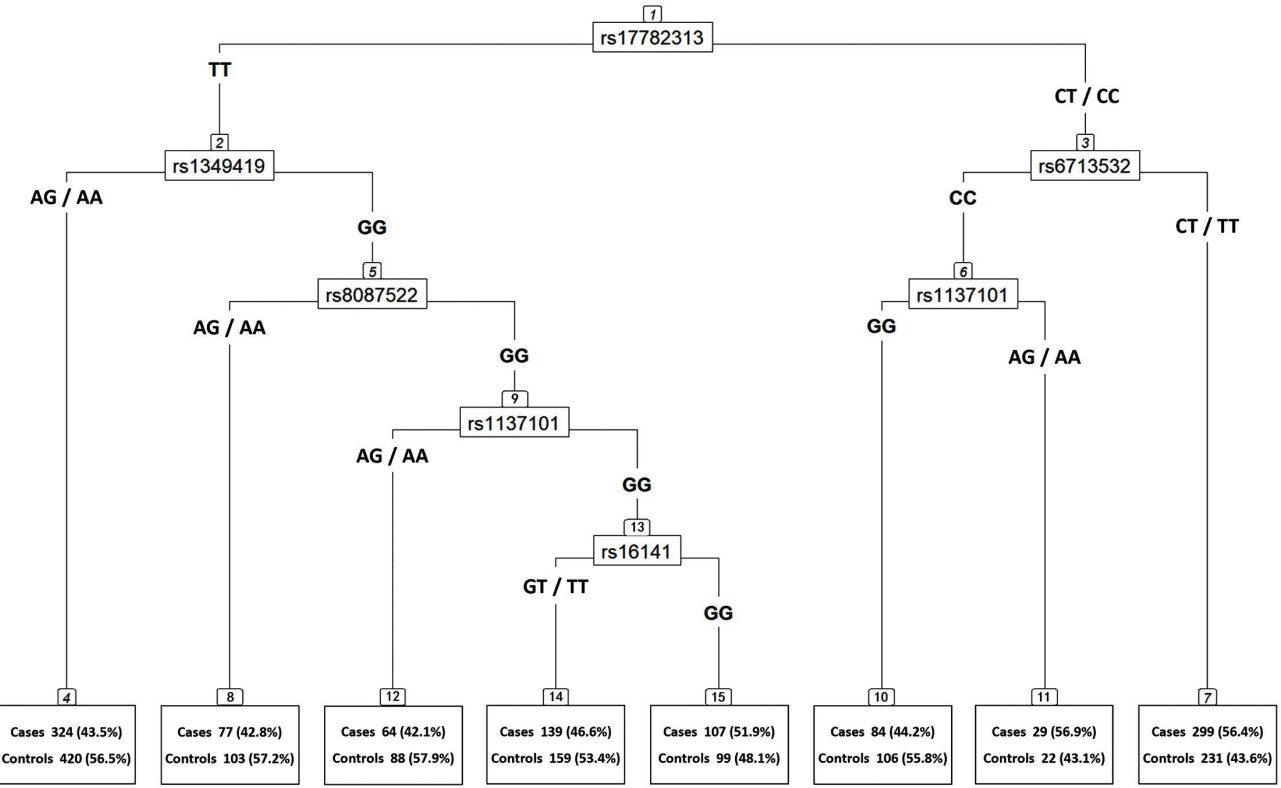

**Fig 2. Classification and regression tree (CART) on genetic variants related to obesity risk in children and adolescents.**

($P_{trend}$<0.001, S1 Fig). The risk of obesity increased by 11% for each risk allele increment (adjusted *OR* = 1.11, 95% *CI* = 1.04–1.18, Table 3). The unweighted GRSs were further categorized as low (0), medium (1−2), and high (3−6). Carriers harboring a high unweighted GRS yielded a 40% greater risk (adjusted *OR* = 1.40, 95% *CI* = 1.09–1.79) compared with those with no risk alleles, while individuals carrying one or two risk alleles did not show an increased obesity risk. The weighted GRSs, ranging from 0 to 1.190, were also categorized as low (0), medium (0.044–0.485), and high (0.486–1.190). Carriers with a high weighted GRS had a 33% higher risk of obesity than those with a low weighted GRS (adjusted *OR* = 1.33, 95% *CI* = 1.04–1.69), but no significant difference was seen between the medium and low weighted-GRS groups (*P* > 0.05).

We further performed stratified analyses by sex and age group. Except for weighted GRS in girls, both high unweighted and weighted GRSs showed significantly positive associations with obesity risk across all subgroups, with no significant differences between sex or age subgroups (all *P* for heterogeneity >0.05, S6 Table). Nevertheless, the effect of high GRS appeared to be more pronounced in the younger group than in the older group.

## Gene–lifestyle interactions on obesity risk

Crossover analysis suggested joint effects of GRSs and lifestyle behaviors on the obesity risk in children and adolescents. As shown in Table 4, individuals with a high unweighted GRS and inadequate sleep had a 70% increased risk of obesity compared with those with a low/medium unweighted GRS and adequate sleep (adjusted *OR*=1.70, 95% *CI* = 1.28–2.26). Similarly, compared with individuals carrying low/medium unweighted GRS and adequate physical activity, those with a high unweighted GRS and inadequate physical activity had a 59% higher *OR* of obesity risk (adjusted *OR*=1.59, 95%

**Table 3. Association between GRS and obesity risk in children and adolescents.**

| GRS[a] | Cases/controls | OR (95% CI) | OR (95% CI)[d] | $P_{trend}$ |
|---|---|---|---|---|
| Unweighted GRS[b] | | | | <0.001 |
| Low (0) | 195/233 | 1.00 | 1.00 | |
| Medium (1–2) | 577/695 | 0.99 (0.80-1.24) | 0.99 (0.79-1.23) | |
| High (3–6) | 350/300 | 1.39 (1.09-1.78) | 1.40 (1.09-1.79) | |
| Low/medium | 772/928 | 1.00 | 1.00 | |
| High | 350/300 | 1.40 (1.17-1.68) | 1.41 (1.17-1.69) | |
| Per one risk allele increment | | 1.11 (1.04-1.18) | 1.11 (1.04-1.18) | |
| Weighted GRS[c] | | | | <0.001 |
| Low (0) | 195/233 | 1.00 | 1.00 | |
| Medium (0.044–0.485) | 567/670 | 1.01 (0.81-1.26) | 1.01 (0.81-1.26) | |
| High (0.486–1.190) | 360/324 | 1.33 (1.04-1.69) | 1.33 (1.04-1.69) | |
| Low/medium | 762/904 | 1.00 | 1.00 | |
| High | 360/324 | 1.32 (1.10-1.58) | 1.32 (1.11-1.58) | |
| Per 0.1 scores increment | | 1.06 (1.03-1.09) | 1.06 (1.03-1.09) | |

CI, confidence interval; GRS, genetic risk score; OR, odds ratio.

[a]GRS data was missing for one case and three controls.

[b]Unweighted GRS was calculated by summing the counts of the risk alleles of four SNPs, including rs17782313 C allele, rs12970134 A allele, rs1137101 A allele, and rs6713532 T allele.

[c]Weighted GRS was calculated by multiplying each $\beta$ coefficient for additive model by the number of the above risk alleles. The $\beta$ coefficients for rs17782313, rs12970134, rs1137101 and rs6713532 were 0.268, 0.217, 0.110 and 0.044, respectively.

[d]Multivariate logistic regression models were adjusted for age, sex, maternal and paternal education levels, and household incomes.

CI = 1.17–2.17). Similar results were observed for the weighted GRS in combination with lifestyle factors (Table 4). However, further analysis showed no statistically significant interactions, with 95% CIs for RERIs and APs including 0 and for IORs including 1.

## Discussion

This case-control study, including 1123 cases with obesity and 1231 controls among children and adolescents, identified *MC4R* rs17782313 and rs12970134, *LEPR* rs1137101, and *POMC* rs6713532 as important genetic variants associated with susceptibility to childhood obesity. More intriguingly, the cumulative effect of these four genetic variants, as well as their joint effect with physical activity and sleep, significantly affect the development of childhood obesity. These findings enrich our understanding of the involvement of the leptin-melanocortin pathway in genetic susceptibility to childhood obesity, while also laying the groundwork for targeted intervention strategies aimed at obesity management for children and adolescents.

Genetic components within the leptin-melanocortin pathway affect the susceptibility to obesity, as these genes play pivotal roles in the maintenance of energy balance [26,27]. GWASs or replication studies in Asian populations have suggested that the *MC4R* variants, rs17782313 and rs12970134, show strong associations with obesity and high BMI [15,28]. Consistent with previous discoveries, our findings from traditional regression analysis indicate that the rs17782313 C allele and the rs12970134 A allele are linked to an increased risk of childhood obesity. Although there is currently no direct functional evidence relating rs17782313 to MC4R expression, bioinformatic analysis indicates that the altered transcriptional control of *MC4R* is the likely functional mechanism. Based on annotation by HaploReg V4.2 [29], rs17782313, involving a T-to-C transition located 188kb downstream of *MC4R*, may alter the regulatory motifs of *MC4R*, and

**Table 4. Interaction analyses between GRS with physical activity and sleep duration on obesity risk in children and adolescents.**

| GRS | Lifestyle factors | Cases/controls | Adjusted OR (95% CI)[a] | RERI (95% CI)[b] | AP (95% CI)[b] | IOR (95% CI)[b] |
|---|---|---|---|---|---|---|
| Unweighted GRS | Sleep duration | | | | | |
| Low/medium | Adequate | 132/181 | 1.00 | | | |
| Low/medium | Inadequate | 627/737 | 1.16 (0.90-1.49) | | | |
| High | Adequate | 72/70 | 1.40 (0.94-2.09) | | | |
| High | Inadequate | 274/221 | 1.70 (1.28-2.26) | 0.14 (−0.55-0.68) | 0.08 (−0.33-0.38) | 1.04 (0.68-1.61) |
| Unweighted GRS | Physical activity | | | | | |
| Low/medium | Adequate | 100/137 | 1.00 | | | |
| Low/medium | Inadequate | 659/780 | 1.16 (0.88-1.54) | | | |
| High | Adequate | 44/34 | 1.83 (1.09-3.07) | | | |
| High | Inadequate | 304/263 | 1.59 (1.17-2.17) | −0.40 (−1.72-0.34) | −0.25 (−0.98-0.23) | 0.75 (0.42-1.31) |
| Weighted GRS | Sleep duration | | | | | |
| Low/medium | Adequate | 137/172 | 1.00 | | | |
| Low/medium | Inadequate | 615/721 | 1.07 (0.83-1.37) | | | |
| High | Adequate | 67/79 | 1.07 (0.72-1.59) | | | |
| High | Inadequate | 286/237 | 1.51 (1.14-2.01) | 0.37 (−0.18-0.83) | 0.25 (−0.12-0.52) | 1.33 (0.86-2.03) |
| Weighted GRS | Physical activity | | | | | |
| Low/medium | Adequate | 95/132 | 1.00 | | | |
| Low/medium | Inadequate | 656/761 | 1.20 (0.90-1.60) | | | |
| High | Adequate | 49/39 | 1.77 (1.08-2.92) | | | |
| High | Inadequate | 307/282 | 1.52 (1.11-2.07) | −0.45 (−1.58-0.25) | −0.30 (−1.01-0.18) | 0.71 (0.42-1.22) |

AP, attributable proportions of interaction; CI, confidence interval; GRS, genetic risk score; IOR, interaction of odds ratio; OR, odds ratio; RERI, relative excess risk of interaction.

[a]Multivariate logistic regression models were adjusted for age, sex, maternal and paternal education levels, and household incomes.

[b]95% CIs of RERI, AP and IOR were estimated by the bootstrap method with 1000 replications.

subsequently affect the binding affinity of the transcription factor, Pou5f1. Additionally, loss of MC4R function is associated with overeating and obesity, and carriers of the rs17782313 C-allele gained more weight than those with the TT homozygote in mice, further supporting the role of this variant in obesity susceptibility [30]. The other variant located downstream of *MC4R*, rs12970134, was scored as category 1f by RegulomeDB (https://regulomedb.org/regulome-search/), providing supporting evidence for its role as an expression quantitative trait locus and its potential impact on transcriptional factor binding and chromatin accessibility, possibly affecting the expression of its target genes. In HaploReg V4.2, the G-to-A variant of rs12970134 was annotated as altering the binding motifs of 12 transcriptional factors, such as the FOXA family, implicated in obesity and other metabolism-related traits. Hence, it is possible that the rs12970134 variant may contribute to obesity susceptibility through its impact on transcriptional activity of *MC4R* expression.

The CART analysis identified *LEPR* rs1137101 and *POMC* rs6713532 as important variants for obesity in addition to the genetic variants in *MC4R*, despite their lack of significance in traditional single-locus analysis, possibly because of the moderate sample size. In terms of biological function, rs1137101 is a non-synonymous SNP in exon 6 of *LEPR*, resulting in the conversion of a glutamine to arginine. As annotated by Polyphen (http://genetics.bwh.harvard.edu/pph2/dbsearch.shtml), the rs1137101 variant likely leads to significant structural or functional changes in the short isoform of *LEPR* but not in the long isoform. A European GWAS suggested a negative correlation of the rs1137101 A-to-G variant with soluble leptin receptor levels [31]. Several genetic variants in high linkage disequilibrium (LD) with rs1137101 are in transcriptional factor binding sites, such as rs10749754 ($R^2 = 0.97$ and D'= 0.99 with rs1137101), which may alter the regulatory motifs of three transcriptional factors, Myf, RREB-1, and TFE. Moreover, this A-to-G variant was correlated with *LEPR* expression

in GTEx (https://www.gtexportal.org/) [32], with a normalized effect size (NES) of 0.21 (*P*<0.001) in brain tissue. This empirical evidence supports the biological rational for the association of rs1137101 with obesity, but still warrants in-depth mechanistic studies.

Another important genetic variant, rs6713532, located in an intron of *POMC*, has been associated with visceral fat and abdominal fat in a European population [33]. *POMC* is an anorexic gene and encodes a pro-peptide that can be cleaved into melanocortin to activate MC4R. HaploReg predicted that this variant yields an enhancer activity cluster and changes the regulatory motif of TATA_disc7. Furthermore, the GTEx portal annotated a positive correlation between the rs6713532 C allele and *POMC* expression, with a NES of 0.26 (*P*<0.001). Nevertheless, few studies have shown evidence for an association of rs6713532 with childhood obesity [34].

We further constructed unweighted and weighted GRSs based on the four important genetic variants, revealing a dose-effect relationship between the GRSs and obesity risk, where obesity risk increases with higher GRSs in children and adolescents. High GRSs, both unweighted and weighted, raised the obesity risk by 33–40% compared with low GRSs. This indicated a cumulative effect of genetic variants from the leptin-melanocortin pathway on susceptibility to obesity, in line with the signaling mechanism of this pathway. These core genetic variants may affect the expression of the *LEPR*, *POMC*, and *MC4R* genes in the hypothalamus, disrupting satiety signaling in the brain and ultimately resulting in excessive energy intake and increased obesity risk.

Apart from the cumulative effect of genetic variants, this study further identified a joint effect of high GRS with both inadequate sleep and physical activity that increases the risk of obesity. However, no statistically significant interactions were detected likely due to limited power. To date, few study has provided conclusive evidence that variants within the leptin–melanocortin pathway interact with either physical activity or sleep duration to influence obesity risk in children and adolescents. Only a handful of investigations have reported, through stratified analyses, that higher levels of physical activity might attenuate the obesogenic effect of *MC4R* variants [15], and adequate sleep duration might buffer the risk conferred by leptin-related polymorphisms [16]. Consequently, the interactive effects of genetic variants in leptin–melanocortin pathway with lifestyle factors on childhood obesity remain to be rigorously tested in larger-scale prospective studies.

Several limitations of this study should be considered. First, the recruitment of participants through the physical monitoring program for new students led to a non-continuous age range in our sample, which may limited the generalization of the current results. Nevertheless, this study has encompassed individuals from childhood to adolescence. Because this study investigates a biological hypothesis, we consider the discontinuity in participant ages to not be of concern. Second, this study may suffer from insufficient statistical power because of moderate sample size, which may overlook interaction between single lifestyle factors and genetic risk. Although this study indicates a joint effect, the interaction lacks statistical significance and should not be overinterpreted as conclusive evidence of a gene-environment interaction. Third, as in any case-control study, the potential for recall bias exists in our study in terms of the data collection on lifestyle factors. Finally, despite efforts to mitigate confounding factors by the use of a traditional adjustment method, the possibility of residual confounding could not be ruled out.

## Conclusion

Our results suggest that rs17782313 and rs12970134 in *MC4R*, rs1137101 in *LEPR*, and rs6713532 in *POMC* are important genetic variants in the leptin-melanocortin pathway for obesity susceptibility in a population of Chinese children. These genetic variants either act independently or in combination through the core leptin-melanocortin signaling pathway. Our findings further indicate that the joint effect of genetic variants in leptin-melanocortin pathway with inadequate sleep and physical activity may contribute to the risk of obesity in children and adolescents. These findings warrant further validation by large-scale prospective studies and exploration of the underlying mechanisms in the future.

 

## Supporting information

**S1 Fig. Trend of increasing risk of obesity with the growing number of risk alleles in the leptin-melanocortin pathway among Chinese children and adolescents.**
(TIF)

**S1 Table. Characteristics of the 12 genetic variants in leptin-melanocortin pathway.**
(DOCX)

**S2 Table. Functional annotations of the 12 genetic variants in leptin-melanocortin pathway.**
(DOCX)

**S3 Table. Genotypic distribution of 12 genetic variants in leptin-melanocortin pathway in cases and controls.**
(DOCX)

**S4 Table. Association between 10 genetic variants in leptin-melanocortin pathway and obesity risk in Chinese children and adolescents.**
(DOCX)

**S5 Table. Obesity risk estimates of the terminal nodes in CART among Chinese children and adolescents.**
(DOCX)

**S6 Table. Association between GRS (high vs. low/medium) and obesity risk in Chinese children and adolescents by sex and age group.**
(DOCX)

**S1 Text. Statistical analytical code.**
(DOCX)

## Acknowledgments

The authors would like to acknowledge all the participants of this study and the staff from the Guangzhou Health Care Promotion Center for Primary and Middle Schools. We thank Catherine Perfect, MA (Cantab), from Liwen Bianji (Edanz) (www.liwenbianji.cn), for editing the English text of a draft of this manuscript.

## Author contributions

**Conceptualization:** Xiaotong Liang, Li Liu.

**Data curation:** Shun Pan, Jiting Ji.

**Formal analysis:** Xiaotong Liang, Yuying Deng, Miao Chen, Ziwei Huang.

**Funding acquisition:** Li Liu.

**Investigation:** Shun Pan, Jiting Ji, Jiaru Deng, Min Chen, Tiantian Xiong, Jianping Liang.

**Methodology:** Xiaotong Liang, Li Liu.

**Project administration:** Jianping Liang, Li Liu.

**Supervision:** Li Liu.

**Validation:** Yuying Deng, Miao Chen, Ziwei Huang.

**Visualization:** Xiaotong Liang.

**Writing – original draft:** Xiaotong Liang.

**Writing – review & editing:** Li Liu.

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
