## [Decision Letter · Decision Letter 0]

1 Feb 2026

PONE-D-25-62126Genetic variants in the Leptin-Melanocortin Pathway and their joint effects with physical activity and sleep duration on risk of childhood obesityPLOS One

Dear Dr. Liu,

Thank you for submitting your manuscript to PLOS ONE. After careful consideration, we feel that it has merit but does not fully meet PLOS ONE’s publication criteria as it currently stands. Therefore, we invite you to submit a revised version of the manuscript that addresses the points raised during the review process. Namely, the article went through thorough revision process performed by 3 reviewers and has also  been carefully evaluated by the  academic editor. All the exposed comments and suggestions have to be taken into consideration and revised accordingly.Please submit your revised manuscript by Mar 18 2026 11:59PM. If you will need more time than this to complete your revisions, please reply to this message or contact the journal office at plosone@plos.org. Please include the following items when submitting your revised manuscript:

A letter that responds to each point raised by all the reviewers. You should upload this letter as a separate file labeled 'Response to Reviewers'.A marked-up copy of your manuscript that highlights changes made to the original version. You should upload this as a separate file labeled 'Revised Manuscript with Track Changes'.An unmarked version of your revised paper without tracked changes. You should upload this as a separate file labeled 'Manuscript'.

We look forward to receiving your revised manuscript.

Kind regards,

Nataša Marčun Varda, PhD

Academic Editor

PLOS One

Journal Requirements:

“This work was supported by the Natural Science Foundation of Guangdong (2023A1515010105).”

3. In the online submission form, you indicated that “The datasets generated and/or analyzed during the current study are not publicly available due to privacy or ethical restrictions but are available from the corresponding author on reasonable request.”

Reviewers' comments:

**Comments to the Author**

1. Is the manuscript technically sound, and do the data support the conclusions?

Reviewer #1: Yes

Reviewer #2: Yes

Reviewer #3: Yes

2. Has the statistical analysis been performed appropriately and rigorously?

Reviewer #1: I Don't Know

Reviewer #2: Yes

Reviewer #3: Yes

3. Have the authors made all data underlying the findings in their manuscript fully available?

Reviewer #1: No

Reviewer #2: Yes

Reviewer #3: No

4. Is the manuscript presented in an intelligible fashion and written in standard English?

Reviewer #1: Yes

Reviewer #2: Yes

Reviewer #3: Yes

5. Review Comments to the Author

Reviewer #1: The case–control study examines genetic variants in the leptin–melanocortin pathway and their associations with childhood obesity in a Chinese population.

The identification of 4 obesity-associated variants and the GRS analyses are supported by existing literature. The application of CART analysis adds value by identifying 2 variants as important contributors despite their lack of statistical significance in single-locus models.

The study did not detect statistically significant additive or multiplicative gene–lifestyle interactions, despite observing joint effects, which contrasts with some larger studies. The authors acknowledge this limitation and attribute it primarily to limited statistical power. Gene–environment interaction analyses typically require larger samples. The self-reported and dichotomised lifestyle variables likely introduce measurement error and reduce power to detect interactions. The cross-sectional design, combined with a retrospective lifestyle assessment, increases susceptibility to recall bias and reverse causation.

Although BMI is considered sufficient for the diagnosis of obesity in children, clinicians should be aware of its limitations. Also, precise, standardised measurement of weight and height using calibrated equipment and proper technique is essential. How did you do “twice averaged”? Please comment.

The non-continuous age range may affect generalizability. Confounding from unmeasured factors such as diet, screen time, and parental obesity remains possible. The restriction to Han Chinese children from Guangzhou further limits generalizability.

The study reinforces the importance of the leptin–melanocortin pathway in childhood obesity. The finding that children with both a high genetic burden and adverse lifestyle factors experience substantially elevated obesity risk has potential relevance for risk stratification and prevention strategies. Modest effect sizes and the cross-sectional design limit clinical translation.

I would suggest clarifying the limitations of interaction analyses more explicitly and avoiding overinterpretation of joint effects as evidence of interaction. I would consider the value of the study primarily in genetic association and GRS findings rather than gene–environment interaction claims.

I recommend that the manuscript be accepted after minor revisions.

Reviewer #2: The study aimed to explore gene-lifestyle interactions between functional genetic variants in leptin melanocortin pathway and obesity in childhood. The authors have constructed genetic risk scores and categorized children into low, medium and high-risk groups for development of obesity based on four important genetic variants and explored the interaction in relation to sleep and exercise habits. The study is very interesting and addresses important modifiable risk factors, which brings the novelty and applicability of the study. However, some sections would benefit with minor revisions.

1. In the introduction section, please consider providing greater detail and mechanism of the proposed interaction between sleep and physical exercise on the expression in the leptin melanocortin pathway (Paragraph 83-90 or in Figure 1).

Methodology is very robust and explained in detail. Results are well presented. Moderate sample size limited the possibility of finding statistically significant interactions between high genetic risk scores and inadequate physical activity and sleep duration, which was emphasized in the discussion.

2. In the discussion section, I suggest an explanation on the lack of significant differences in physical activity and sleep duration between cases and controls (paragraph 195-199).

The Conclusion section is concise and addresses future directions.

Reviewer #3: The authors have conducted and written a thorough study on the effect of genetic risk scores derived from variants on the leptin-melanocortin pathway and their effects on childhood obesity, as well as the interaction between the genetic risk scores with some lifestyle behaviors (sleep and physical activity) on the risk of obesity. The study strengths include a frequency-matched case control design, rigorous statistics and a robust sample size. They genotyped twelve potentially functional variants and identified four of them as being associated with childhood obesity, and constructed a GRS using those variants. They found that a high unweighted and weighted GRS using those variants conferred a high risk of childhood obesity, with a joint effect from poor physical activity or sleep.

My main comment pertains to whether the authors may be able to explain at least briefly in the manuscript why they decided to focus specifically on physical activity and sleep and why other lifestyle factors such as dietary quality for example were not included.

My second comment pertains to data availability. Since the authors selected that some restrictions apply, would they be able to describe in detail what these restrictions are?

Minor issues were a few small grammatical errors found throughout the text, would recommend a thorough check for these (examples: line 252, line 314, line 319).

6. PLOS authors have the option to publish the peer review history of their article (what does this mean?). If published, this will include your full peer review and any attached files.

Reviewer #1: No

Reviewer #2: No

Reviewer #3: No

---

## [Author Response · Author response to Decision Letter 1]

10 Apr 2026

Responses to editor

Comment 1. Please ensure that your manuscript meets PLOS ONE's style requirements, including those for file naming.

Response: Thank you for your comment. We have carefully checked and revised our manuscript to ensure it complies with PLOS ONE’s style requirements, including the correct file naming format.

Comment 2: Thank you for stating the following financial disclosure: “This work was supported by the Natural Science Foundation of Guangdong (2023A1515010105).” Please state what role the funders took in the study. If the funders had no role, please state: “The funders had no role in study design, data collection and analysis, decision to publish, or preparation of the manuscript.” If this statement is not correct you must amend it as needed. Please include this amended Role of Funder statement in your cover letter; we will change the online submission form on your behalf.

Response: We have added the following statement in the funding section: “The funders had no role in study design, data collection and analysis, decision to publish, or preparation of the manuscript.”

Comment 3: In the online submission form, you indicated that “The datasets generated and/or analyzed during the current study are not publicly available due to privacy or ethical restrictions but are available from the corresponding author on reasonable request.” All PLOS journals now require all data underlying the findings described in their manuscript to be freely available to other researchers, either 1. In a public repository, 2. Within the manuscript itself, or 3. Uploaded as supplementary information. This policy applies to all data except where public deposition would breach compliance with the protocol approved by your research ethics board. If your data cannot be made publicly available for ethical or legal reasons (e.g., public availability would compromise patient privacy), please explain your reasons on resubmission and your exemption request will be escalated for approval.

Response: Thank you for clarifying the data availability policy. We fully understand and respect the requirement for public data sharing. According to the requirement of the Regulation on the Administration of Human Genetic Resource of China and the ethnical approval documents of this study, individual raw genetic data are not publicly available due to ethnical constrains and privacy protection requirements. Any request for individual raw data will be reviewed and approved by Ethnics Committee before access is granted. Additionally, we have provided the full original statistical analytical code and records as supplementary materials to ensure transparency and reproducibility of this study.

Comment 4. When completing the data availability statement of the submission form, you indicated that you will make your data available on acceptance. We strongly recommend all authors decide on a data sharing plan before acceptance, as the process can be lengthy and hold up publication timelines. Please note that, though access restrictions are acceptable now, your entire data will need to be made freely accessible if your manuscript is accepted for publication. This policy applies to all data except where public deposition would breach compliance with the protocol approved by your research ethics board. If you are unable to adhere to our open data policy, please kindly revise your statement to explain your reasoning and we will seek the editor's input on an exemption. Please be assured that, once you have provided your new statement, the assessment of your exemption will not hold up the peer review process.

Response: We have revised our data availability statement as follow: “Summary data are included in the manuscript and supplementary information. Individual raw data are not publicly available because of participant privacy protection, ethical restrictions, and the Regulation on the Administration of Human Genetic Resources of China. Access to individual participant data may be granted to other researchers only after approval from the Ethics Committee of Guangdong Pharmaceutical University (contact via E-mail: gylunli@163.net). The original statistical analysis code and programming log of this study are provided as supplementary materials (S1 Text)”.

Comment 5. Please include captions for your Supporting Information files at the end of your manuscript, and update any in-text citations to match accordingly. Please see our Supporting Information guidelines for more information: http://journals.plos.org/plosone/s/supporting-information

Response: We have included captions for our Supporting Information files at the end of our manuscript.

Comment 6. If the reviewer comments include a recommendation to cite specific previously published works, please review and evaluate these publications to determine whether they are relevant and should be cited. There is no requirement to cite these works unless the editor has indicated otherwise.

Response: Thank you for your reminder, wo have reviewed and revised manuscript according the suggestions.

Comment 7. Please review your reference list to ensure that it is complete and correct. If you have cited papers that have been retracted, please include the rationale for doing so in the manuscript text, or remove these references and replace them with relevant current references. Any changes to the reference list should be mentioned in the rebuttal letter that accompanies your revised manuscript. If you need to cite a retracted article, indicate the article’s retracted status in the References list and also include a citation and full reference for the retraction notice.

Response: We have checked the references to ensure completeness and accuracy. No retracted articles were cited in our manuscript.

Response to Reviewer #1:

Comment 1. The case–control study examines genetic variants in the leptin–melanocortin pathway and their associations with childhood obesity in a Chinese population.

The identification of 4 obesity-associated variants and the GRS analyses are supported by existing literature. The application of CART analysis adds value by identifying 2 variants as important contributors despite their lack of statistical significance in single-locus models.

The study did not detect statistically significant additive or multiplicative gene–lifestyle interactions, despite observing joint effects, which contrasts with some larger studies. The authors acknowledge this limitation and attribute it primarily to limited statistical power. Gene–environment interaction analyses typically require larger samples. The self-reported and dichotomised lifestyle variables likely introduce measurement error and reduce power to detect interactions. The cross-sectional design, combined with a retrospective lifestyle assessment, increases susceptibility to recall bias and reverse causation. Although BMI is considered sufficient for the diagnosis of obesity in children, clinicians should be aware of its limitations. Also, precise, standardised measurement of weight and height using calibrated equipment and proper technique is essential. How did you do “twice averaged”? Please comment.

Response: Thank you for the positive comments and suggestions. We have added the methodological details in Method section: “ Height and weight were measured by trained technicians using standardized protocols and calibrated equipment. Height was measured to the nearest 0.1 cm using a mechanical height gauge, and weight was measured to the nearest 0.1 kg using a lever scale. For each participants, both measurements were done twice in light clothing without shoes, and the average of each measurement was calculated. BMI was calculated as weight (kg) divided by height (m) squared.”

Comment 2. The non-continuous age range may affect generalizability. Confounding from unmeasured factors such as diet, screen time, and parental obesity remains possible. The restriction to Han Chinese children from Guangzhou further limits generalizability. The study reinforces the importance of the leptin–melanocortin pathway in childhood obesity. The finding that children with both a high genetic burden and adverse lifestyle factors experience substantially elevated obesity risk has potential relevance for risk stratification and prevention strategies. Modest effect sizes and the cross-sectional design limit clinical translation. I would suggest clarifying the limitations of interaction analyses more explicitly and avoiding overinterpretation of joint effects as evidence of interaction. I would consider the value of the study primarily in genetic association and GRS findings rather than gene–environment interaction claims. I recommend that the manuscript be accepted after minor revisions.

Response: We sincerely appreciate the valuable suggestion. To be more cautions, we have revised the Discussion section: “… Second, this study may suffer from insufficient statistical power because of moderate sample size, which may overlooked interaction between single lifestyle factors and genetic risk. Although this study indicates a joint effect, the interaction lacks statistical significance and should not be overinterpreted as conclusive evidence of a gene-environment interaction. …”

Response to Reviewer #2:

Comment 1. The study aimed to explore gene-lifestyle interactions between functional genetic variants in leptin melanocortin pathway and obesity in childhood. The authors have constructed genetic risk scores and categorized children into low, medium and high-risk groups for development of obesity based on four important genetic variants and explored the interaction in relation to sleep and exercise habits. The study is very interesting and addresses important modifiable risk factors, which brings the novelty and applicability of the study. However, some sections would benefit with minor revisions. In the introduction section, please consider providing greater detail and mechanism of the proposed interaction between sleep and physical exercise on the expression in the leptin melanocortin pathway (Paragraph 83-90 or in Figure 1).Methodology is very robust and explained in detail. Results are well presented. Moderate sample size limited the possibility of finding statistically significant interactions between high genetic risk scores and inadequate physical activity and sleep duration, which was emphasized in the discussion.

Response: Thank you very much for your positive comments and constructive suggestions. We have revised the Introduction section: “Exercise and sufficient sleep could improve leptin resistance and promote leptin signaling [14]. Current studies have mainly concentrated on how single genetic variants in the leptin pathway interact with some behavioral factors, such as MC4R rs12970134 in interaction with physical activity [15], or the interaction between leptin-related polygenic risk and short sleep duration in modifying childhood obesity susceptibility [16].”

Comment 2. In the discussion section, I suggest an explanation on the lack of significant differences in physical activity and sleep duration between cases and controls (paragraph 195-199). The Conclusion section is concise and addresses future directions.

Response: Thank you for the positive comment and suggestion. Several reasons may explain the lack of significant differences in physical activity and sleep duration between cases and controls. First, the number of participants with adequate physical activity or adequate sleep duration was relatively small, which may reduce the statistical power to detect differences. Second, retrospective lifestyle assessment in the case-control study may be susceptible to recall bias. These limitations have been stated in the revised discussion section. Further large-sample prospective studies are warranted to verify and extend our findings.

Response to Reviewer #3:

Comment 1: The authors have conducted and written a thorough study on the effect of genetic risk scores derived from variants on the leptin-melanocortin pathway and their effects on childhood obesity, as well as the interaction between the genetic risk scores with some lifestyle behaviors (sleep and physical activity) on the risk of obesity. The study strengths include a frequency-matched case control design, rigorous statistics and a robust sample size. They genotyped twelve potentially functional variants and identified four of them as being associated with childhood obesity, and constructed a GRS using those variants. They found that a high unweighted and weighted GRS using those variants conferred a high risk of childhood obesity, with a joint effect from poor physical activity or sleep. My main comment pertains to whether the authors may be able to explain at least briefly in the manuscript why they decided to focus specifically on physical activity and sleep and why other lifestyle factors such as dietary quality for example were not included.

Response: Thank you very much for your positive comment and valuable suggestion. Physical activity and sleep were selected mainly because they are closely related to the regulation of leptin signaling. In addition, we did not collect sufficient data on dietary quality in this current study. Therefore, we focused on physical activity and sleep. We have added a brief explanation in the Introduction section: “Exercise and sufficient sleep could improve leptin resistance and promote leptin signaling [14]. Current studies have mainly concentrated on how single genetic variants in the leptin pathway interact with some behavioral factors, such as MC4R rs12970134 in interaction with physical activity [15], or the interaction between leptin-related polygenic risk and short sleep duration in modifying childhood obesity susceptibility [16].”

Comment 2. My second comment pertains to data availability. Since the authors selected that some restrictions apply, would they be able to describe in detail what these restrictions are?

Response: Thank you for the comment. According to the requirement of the Regulation on the Administration of Human Genetic Resource of China and the ethnical approval documents of this study, individual raw genetic data are not publicly available due to ethnical constrains and privacy protection requirements. Any request for individual raw data will be reviewed and approved by Ethnics Committee before access is granted. We have provided the full original statistical analytical code and records as supplementary materials to ensure transparency and reproducibility of this study. We have revised our data availability statement as follow: “Summary data are included in the manuscript and supplementary information. Individual raw data are not publicly available because of participant privacy protection, ethical restrictions, and the Regulation on the Administration of Human Genetic Resources of China. Access to individual participant data may be granted to other researchers only after approval from the Ethics Committee of Guangdong Pharmaceutical University (contact via E-mail: gylunli@163.net). The original statistical analysis code and programming log of this study are provided as supplementary materials (S1 Text)”.

Comment 3. Minor issues were a few small grammatical errors found throughout the text, would recommend a thorough check for these (examples: line 252, line 314, line 319).

Response: Thank you for your careful review and helpful suggestion. We have carefully checked and revised the entire manuscript for grammatical errors.

---

## [Decision Letter · Decision Letter 1]

20 Apr 2026

Genetic variants in the Leptin-Melanocortin Pathway and their joint effects with physical activity and sleep duration on risk of childhood obesity

PONE-D-25-62126R1

Dear Dr. Li Liu,

we re pleased to inform you that your manuscript has been judged scientifically suitable for publication and will be formally accepted for publication once it meets all outstanding technical requirements.

Within one week, you’ll receive an e-mail detailing the required amendments. When these have been addressed, you’ll receive a formal acceptance letter and your manuscript will be scheduled for publication. Attention has to be paid to the data availability upon reasonable request.

Kind regards,

Nataša Marčun Varda, PhD

Academic Editor

PLOS One

Reviewers' comments:

Reviewer's Responses to Questions

**Comments to the Author**

1. If the authors have adequately addressed your comments raised in a previous round of review and you feel that this manuscript is now acceptable for publication, you may indicate that here to bypass the “Comments to the Author” section, enter your conflict of interest statement in the “Confidential to Editor” section, and submit your "Accept" recommendation.

Reviewer #1: All comments have been addressed

Reviewer #2: All comments have been addressed

2. Is the manuscript technically sound, and do the data support the conclusions?

Reviewer #1: Yes

Reviewer #2: Yes

3. Has the statistical analysis been performed appropriately and rigorously?

Reviewer #1: Yes

Reviewer #2: Yes

4. Have the authors made all data underlying the findings in their manuscript fully available?

Reviewer #1: (No Response)

Reviewer #2: No

5. Is the manuscript presented in an intelligible fashion and written in standard English?

Reviewer #1: Yes

Reviewer #2: Yes

6. Review Comments to the Author

Reviewer #1: (No Response)

Reviewer #2: Comment on Q4: Individual-level raw data are not publicly available due to participant privacy considerations, ethical restrictions, and the Regulation on the Administration of Human Genetic Resources of China. Data may be available upon reasonable request via email.

7. PLOS authors have the option to publish the peer review history of their article (what does this mean?). If published, this will include your full peer review and any attached files.

Reviewer #1: No

Reviewer #2: No

---

## [Editor Report · Acceptance letter]

PONE-D-25-62126R1

PLOS One

Dear Dr. Liu,

I'm pleased to inform you that your manuscript has been deemed suitable for publication in PLOS One. Congratulations! Your manuscript is now being handed over to our production team.

Kind regards,

on behalf of

Prof. Dr. Nataša Marčun Varda

Academic Editor

PLOS One